# A Neural-Symbolic Approach to Natural Language Tasks

## Abstract

Deep learning (DL) has in recent years been widely used in natural language processing (NLP) applications due to its superior performance. However, while natural languages are rich in grammatical structure, DL does not have an internal representation to explicitly represent and enforce such structures. This paper proposes a new architecture to bridge this gap by exploiting tensor product representations (TPR), a structured neural-symbolic framework developed in cognitive science over the past 20 years, with the aim of integrating DL with explicit language structures and rules. We call it the *Tensor Product Generation Network* (**TPGN**), and apply it to image captioning. The key ideas of TPGN are: 1) unsupervised learning of *role-unbinding vectors* of words via a TPR-based deep neural network, and 2) integration of TPR with typical DL architectures including Long Short-Term Memory (LSTM) models. The novelty of our approach lies in its ability to generate a sentence and extract partial grammatical structure of the sentence by using role-unbinding vectors, which are obtained in an unsupervised manner. Experimental results demonstrate the effectiveness of the proposed approach.

## 1 Introduction

Deep learning is an important tool in many current natural language processing (NLP) applications. However, language rules or structures cannot be explicitly represented in deep learning architectures. The tensor product representation developed in Smolensky (1990); Smolensky & Legendre (2006) has the potential of integrating deep learning with explicit rules (such as logical rules, grammar rules, or rules that summarize real-world knowledge). This paper develops a TPR approach for deep-learning-based NLP applications, introducing the *Tensor Product Generation Network (TPGN)* architecture. To demonstrate the effectiveness of the proposed architecture, we apply it to a important NLP application: image captioning.

A TPGN model generates natural language descriptions via learned representations. The representations learned in the TPGN can be interpreted as encoding grammatical roles for the words being generated. This layer corresponds to the role-encoding component of a general, independently-developed architecture for neural computation of symbolic functions, including the generation of linguistic structures. The key to this architecture is the notion of *Tensor Product Representation (TPR)*, in which vectors embedding symbols (e.g., `lives`, `frodo`) are bound to vectors embedding structural roles (e.g., `verb`, `subject`) and combined to generate vectors embedding symbol structures (`[frodo lives]`). TPRs provide the representational foundations for a general computational architecture called *Gradient Symbolic Computation (GSC)*, and applying GSC to the task of natural language generation yields the specialized architecture defining the model presented here. The generality of GSC means that the results reported here have implications well beyond the particular tasks we address here.

The paper is organized as follows. Section 2 discusses related work. In Section 3, we review the basics of tensor product representation. Section 4 presents the rationale for our proposed architecture. Section 5 describes our proposed model in detail. In Section 6, we present our experimental results. Finally, Section 7 concludes the paper.

## 2 RELATED WORK

Deep learning plays a dominant role in many NLP applications due to its exceptional performance. Hence, we focus on recent deep-learning-based literature for an important NLP application, i.e., image captioning.

This work follows a great deal of recent caption-generation literature in exploiting end-to-end deep learning with a CNN image-analysis front end producing a distributed representation that is then used to drive a natural-language generation process, typically using RNNs Mao et al. (2015); Vinyals et al. (2015); Devlin et al. (2015); Chen & Zitnick (2015); Donahue et al. (2015); Karpathy & Fei-Fei (2015); Kiros et al. (2014a;b); Fang et al. (2015). Our grammatical interpretation of the structural roles of words in sentences makes contact with other work that incorporates deep learning into grammatically-structured networks Tai et al. (2015); Kumar et al. (2016); Kong et al. (2017); Andreas et al. (2015); Yogatama et al. (2016); Maillard et al. (2017); Socher et al. (2010); Pollack (1990). Here, the network is not itself structured to match the grammatical structure of sentences being processed; the structure is fixed, but is designed to support the learning of distributed representations that incorporate structure internal to the representations themselves — filler/role structure.

TPRs are also used in NLP in Palangi et al. (2017) but there the representation of each individual input word is constrained to be a literal TPR filler/role binding. (The idea of using the outer product to construct internal representations was also explored in Fukui et al. (2016).) Here, by contrast, the learned representations are not themselves constrained, but the global structure of the network is designed to display the somewhat abstract property of being TPR-capable: the architecture uses the TPR unbinding operation of the matrix-vector product to extract individual words for sequential output.

## 3 REVIEW OF TENSOR PRODUCT REPRESENTATION

Tensor product representation (TPR) is a general framework for embedding a space of symbol structures $\mathfrak{S}$ into a vector space. This embedding enables neural network operations to perform symbolic computation, including computations that provide considerable power to symbolic NLP systems (Smolensky & Legendre (2006); Smolensky (2012)). Motivated by these successful examples, we are inspired to extend the TPR to the challenging task of learning image captioning. And as a by-product, the symbolic character of TPRs makes them amenable to conceptual interpretation in a way that standard learned neural network representations are not.

A particular TPR embedding is based in a *filler/role decomposition* of $\mathfrak{S}$. A relevant example is when $\mathfrak{S}$ is the set of strings over an alphabet $\{\mathtt{a}, \mathtt{b}, \ldots\}$. One filler/role decomposition deploys the *positional roles* $\{r_k\}, k \in \mathbb{N}$, where the *filler/role binding* $\mathtt{a}/r_k$ assigns the 'filler' (symbol) $\mathtt{a}$ to the $k^{th}$ position in the string. A string such as $\mathtt{abc}$ is uniquely determined by its filler/role bindings, which comprise the (unordered) set $\mathfrak{B}(\mathtt{abc}) = \{\mathtt{b}/r_2, \mathtt{a}/r_1, \mathtt{c}/r_3\}$. Reifying the notion *role* in this way is key to TPR's ability to encode complex symbol *structures*.

Given a selected filler/role decomposition of the symbol space, a particular TPR is determined by an embedding that assigns to each filler a vector in a vector space $V_F \cong \mathbb{R}^{d_F}$, and a second embedding that assigns to each role a vector in a space $V_R \cong \mathbb{R}^{d_R}$. The vector embedding a symbol $\mathtt{a}$ is denoted by $\mathbf{f}_\mathtt{a}$ and is called a *filler vector*; the vector embedding a role $r_k$ is $\mathbf{r_k}$ and called a *role vector*. The TPR for $\mathtt{abc}$ is then the following 2-index tensor in $V_F \otimes V_R \cong \mathbb{R}^{d_F \times d_R}$:

$$\mathbf{S}_\mathtt{abc} = \mathbf{f}_\mathtt{b} \otimes \mathbf{r_2} + \mathbf{f}_\mathtt{a} \otimes \mathbf{r_1} + \mathbf{f}_\mathtt{c} \otimes \mathbf{r_3}, \tag{1}$$

where $\otimes$ denotes the tensor product. The tensor product is a generalization of the vector outer product that is recursive; recursion is exploited in TPRs for, e.g., the distributed representation of trees, the neural encoding of formal grammars in connection weights, and the theory of neural computation of recursive symbolic functions. Here, however, it suffices to use the outer product; using matrix notation we can write (1) as:

$$\mathbf{S}_\mathtt{abc} = \mathbf{f}_\mathtt{b}\mathbf{r_2}^\top + \mathbf{f}_\mathtt{a}\mathbf{r_1}^\top + \mathbf{f}_\mathtt{c}\mathbf{r_3}^\top. \tag{2}$$

Generally, the embedding of any symbol structure $\mathtt{S} \in \mathfrak{S}$ is $\sum\{\mathbf{f}_i \otimes \mathbf{r}_i \mid \mathtt{f}_i/r_i \in \mathfrak{B}(\mathtt{S})\}$; here: $\sum\{\mathbf{f}_i\mathbf{r}_i^\top \mid \mathtt{f}_i/r_i \in \mathfrak{B}(\mathtt{S})\}$ (Smolensky (1990); Smolensky & Legendre (2006)).

A key operation on TPRs, central to the work presented here, is *unbinding*, which undoes binding. Given the TPR in (2), for example, we can unbind $\mathbf{r_2}$ to get $\mathbf{f_b}$; this is achieved simply by $\mathbf{f_b} = \mathbf{S}_{\text{abc}}\mathbf{u_2}$. Here $\mathbf{u_2}$ is *the unbinding vector dual to* the binding vector $\mathbf{r_2}$. To make such exact unbinding possible, the role vectors should be chosen to be linearly independent. (In that case the unbinding vectors are the rows of the inverse of the matrix containing the binding vectors as columns, so that $\mathbf{r_2} \cdot \mathbf{u_2} = 1$ while $\mathbf{r_k} \cdot \mathbf{u_2} = 0$ for all other role vectors $\mathbf{r_k} \neq \mathbf{r_2}$; this entails that $\mathbf{S}_{\text{abc}}\mathbf{u_2} = \mathbf{b}$, the filler vector bound to $\mathbf{r_2}$. Replacing the matrix inverse with the pseudo-inverse allows approximate unbinding when the role vectors are not linearly independent).

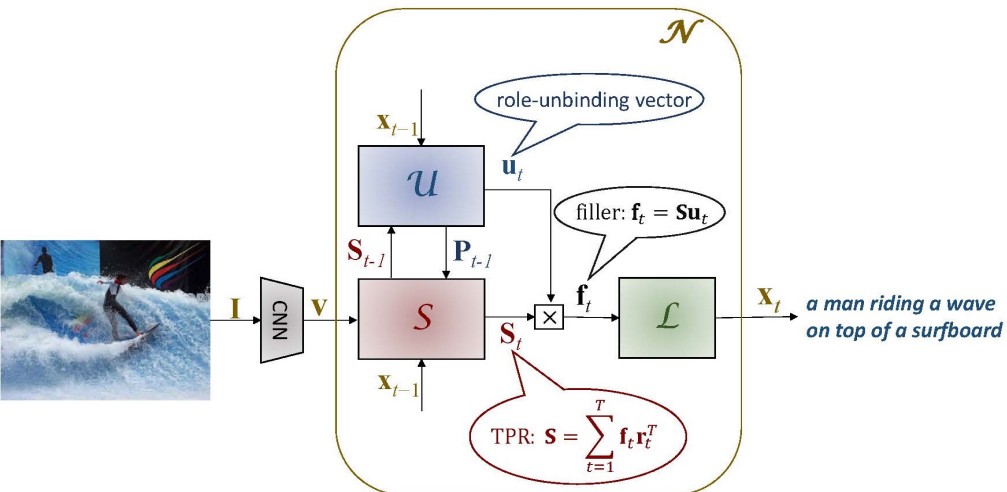

Figure 1: Architecture of TPGN, a TPR-capable generation network. "⊠" denotes the matrix-vector product.

## 4 A TPR-CAPABLE GENERATION ARCHITECTURE

In this work we propose an approach to network architecture design we call the *TPR-capable method*. The architecture we use (see Fig. 1) is designed so that TPRs could, in theory, be used within the architecture to perform the target task — here, generating a caption one word at a time. Unlike previous work where TPRs are hand-crafted, in our work, end-to-end deep learning will induce representations which the architecture can use to generate captions effectively.

In this section, we consider the problem of image captioning. As shown in Fig. 1, our proposed system is denoted by $\mathcal{N}$, which is from "N" in "TPGN". The input of $\mathcal{N}$ is an image feature vector $\mathbf{v}$ and the output of $\mathcal{N}$ is a caption. The image feature vector $\mathbf{v}$ is extracted from a given image by a pre-trained CNN. The first part of our system $\mathcal{N}$ is a *sentence-encoding subnetwork* $\mathcal{S}$ which maps $\mathbf{v}$ to a representation $S$ which will drive the entire caption-generation process; $S$ contains all the image-specific information for producing the caption. (We will call a caption a "sentence" even though it may in fact be just a noun phrase.)

If $S$ were a TPR of the caption itself, it would be a matrix (or 2-index tensor) $\mathbf{S}$ which is a sum of matrices, each of which encodes the binding of one word to its role in the sentence constituting the caption. To serially read out the words encoded in $\mathbf{S}$, in iteration 1 we would *unbind* the first word from $\mathbf{S}$, then in iteration 2 the second, and so on. As each word is generated, $\mathbf{S}$ could update itself, for example, by subtracting out the contribution made to it by the word just generated; $\mathbf{S}_t$ denotes the value of $\mathbf{S}$ when word $w_t$ is generated. At time step $t$ we would unbind the role $r_t$ occupied by word $w_t$ of the caption. So the second part of our system $\mathcal{N}$ — the *unbinding subnetwork* $\mathcal{U}$ — would generate, at iteration $t$, the *unbinding vector* $\mathbf{u}_t$. Once $\mathcal{U}$ produces the unbinding vector $\mathbf{u}_t$, this vector would then be applied to $\mathbf{S}$ to extract the symbol $\mathbf{f_t}$ that occupies word $t$'s role; the symbol represented by $\mathbf{f_t}$ would then be decoded into word $w_t$ by the third part of $\mathcal{N}$, i.e., the *lexical decoding subnetwork* $\mathcal{L}$, which outputs $\mathbf{x}_t$, the 1-hot-vector encoding of $w_t$.

Recalling that unbinding in TPR is achieved by the matrix-vector product, the key operation in generating $w_t$ is thus the unbinding of $r_t$ within $\mathbf{S}$, which amounts to simply:

$$\mathbf{S}_t \mathbf{u_t} = \mathbf{f_t}. \tag{3}$$

This matrix-vector product is denoted "⊠" in Fig. 1.

Thus the system $\mathcal{N}$ of Fig. 1is TPR-capable. This is what we propose as the Tensor-Product Generation Network (TPGN) architecture. The learned representation $\mathbf{S}$ will not be proven to literally be a TPR, but by analyzing the unbinding vectors $\mathbf{u}_t$ the network learns, we will gain insight into the process by which the learned matrix $\mathbf{S}$ gives rise to the generated caption.

What type of roles might the unbinding vectors be unbinding? A TPR for a caption could in principle be built upon *positional roles*, *syntactic/semantic roles*, or some combination of the two. In the caption ***a man standing in a room with a suitcase***, the initial *a* and *man* might respectively occupy the positional roles of POS(ITION)$_1$ and POS$_2$; *standing* might occupy the syntactic role of VERB; *in* the role of SPATIAL-P(REPOSITION); while *a room with a suitcase* might fill a 5-role schema DET(ERMINER)$_1$ N(OUN)$_1$ P DET$_2$ N$_2$. In fact we will see evidence below that our network learns just this kind of hybrid role decomposition.

What form of information does the sentence-encoding subnetwork $\mathcal{S}$ need to encode in $\mathbf{S}$? Continuing with the example of the previous paragraph, $\mathbf{S}$ needs to be some approximation to the TPR summing several filler/role binding matrices. In one of these bindings, a filler vector $\mathbf{f}_a$ — which the lexical subnetwork $\mathcal{L}$ will map to the article *a* — is bound (via the outer product) to a role vector $\mathbf{r}_{\text{POS}_1}$ which is the dual of the first unbinding vector produced by the unbinding subnetwork $\mathcal{U}$: $\mathbf{u}_{\text{POS}_1}$. In the first iteration of generation the model computes $\mathbf{S}_1 \mathbf{u}_{\text{POS}_1} = \mathbf{f}_a$, which $\mathcal{L}$ then maps to *a*. Analogously, another binding approximately contained in $\mathbf{S}_2$ is $\mathbf{f}_{man}\mathbf{r}_{\text{POS}_2}^\top$. There are corresponding bindings for the remaining words of the caption; these employ syntactic/semantic roles. One example is $\mathbf{f}_{standing}\mathbf{r}_V^\top$. At iteration 3, $\mathcal{U}$ decides the next word should be a verb, so it generates the unbinding vector $\mathbf{u}_V$ which when multiplied by the current output of $\mathcal{S}$, the matrix $\mathbf{S}_3$, yields a filler vector $\mathbf{f}_{standing}$ which $\mathcal{L}$ maps to the output *standing*. $\mathcal{S}$ decided the caption should deploy *standing* as a verb and included in $\mathbf{S}$ the binding $\mathbf{f}_{standing}\mathbf{r}_V^\top$. It similarly decided the caption should deploy *in* as a spatial preposition, including in $\mathbf{S}$ the binding $\mathbf{f}_{in}\mathbf{r}_{\text{SPATIAL-P}}^\top$; and so on for the other words in their respective roles in the caption.

## 5 SYSTEM DESCRIPTION

The unbinding subnetwork $\mathcal{U}$ and the sentence-encoding network $\mathcal{S}$ of Fig. 1 are each implemented as (1-layer, 1-directional) LSTMs (see Fig. 2); the lexical subnetwork $\mathcal{L}$ is implemented as a linear transformation followed by a softmax operation. In the equations below, the LSTM variables internal to the $\mathcal{S}$ subnet are indexed by 1 (e.g., the forget-, input-, and output-gates are respectively $\hat{\mathbf{f}}_1, \hat{\mathbf{i}}_1, \hat{\mathbf{o}}_1$) while those of the unbinding subnet $\mathcal{U}$ are indexed by 2.

Thus the state updating equations for $\mathcal{S}$ are, for $t = 1, \cdots, T =$ caption length:

$$\hat{\mathbf{f}}_{1,t} = \sigma_g(\mathbf{W}_{1,f}\mathbf{p}_{t-1} - \mathbf{D}_{1,f}\mathbf{W}_e\mathbf{x}_{t-1} + \mathbf{U}_{1,f}\hat{\mathbf{S}}_{t-1}) \tag{4}$$

$$\hat{\mathbf{i}}_{1,t} = \sigma_g(\mathbf{W}_{1,i}\mathbf{p}_{t-1} - \mathbf{D}_{1,i}\mathbf{W}_e\mathbf{x}_{t-1} + \mathbf{U}_{1,i}\hat{\mathbf{S}}_{t-1}) \tag{5}$$

$$\hat{\mathbf{o}}_{1,t} = \sigma_g(\mathbf{W}_{1,o}\mathbf{p}_{t-1} - \mathbf{D}_{1,o}\mathbf{W}_e\mathbf{x}_{t-1} + \mathbf{U}_{1,o}\hat{\mathbf{S}}_{t-1}) \tag{6}$$

$$\mathbf{g}_{1,t} = \sigma_h(\mathbf{W}_{1,c}\mathbf{p}_{t-1} - \mathbf{D}_{1,c}\mathbf{W}_e\mathbf{x}_{t-1} + \mathbf{U}_{1,c}\hat{\mathbf{S}}_{t-1}) \tag{7}$$

$$\mathbf{c}_{1,t} = \hat{\mathbf{f}}_{1,t} \odot \mathbf{c}_{1,t-1} + \hat{\mathbf{i}}_{1,t} \odot \mathbf{g}_{1,t} \tag{8}$$

$$\hat{\mathbf{S}}_t = \hat{\mathbf{o}}_{1,t} \odot \sigma_h(\mathbf{c}_{1,t}) \tag{9}$$

where $\hat{\mathbf{f}}_{1,t}, \hat{\mathbf{i}}_{1,t}, \hat{\mathbf{o}}_{1,t}, \mathbf{g}_{1,t}, \mathbf{c}_{1,t}, \hat{\mathbf{S}}_t \in \mathbb{R}^{d \times d}$, $\mathbf{p}_t \in \mathbb{R}^d$, $\sigma_g(\cdot)$ is the (element-wise) logistic sigmoid function; $\sigma_h(\cdot)$ is the hyperbolic tangent function; the operator $\odot$ denotes the Hadamard (element-wise) product; $\mathbf{W}_{1,f}, \mathbf{W}_{1,i}, \mathbf{W}_{1,o}, \mathbf{W}_{1,c} \in \mathbb{R}^{d \times d \times d}$, $\mathbf{D}_{1,f}, \mathbf{D}_{1,i}, \mathbf{D}_{1,o}, \mathbf{D}_{1,c} \in \mathbb{R}^{d \times d \times d}$, $\mathbf{U}_{1,f}, \mathbf{U}_{1,i}, \mathbf{U}_{1,o}, \mathbf{U}_{1,c} \in \mathbb{R}^{d \times d \times d \times d}$. For clarity, biases — included throughout the model — are omitted from all equations in this paper. The initial state $\hat{\mathbf{S}}_0$ is initialized by:

$$\hat{\mathbf{S}}_0 = \mathbf{C}_s(\mathbf{v} - \bar{\mathbf{v}}) \tag{10}$$

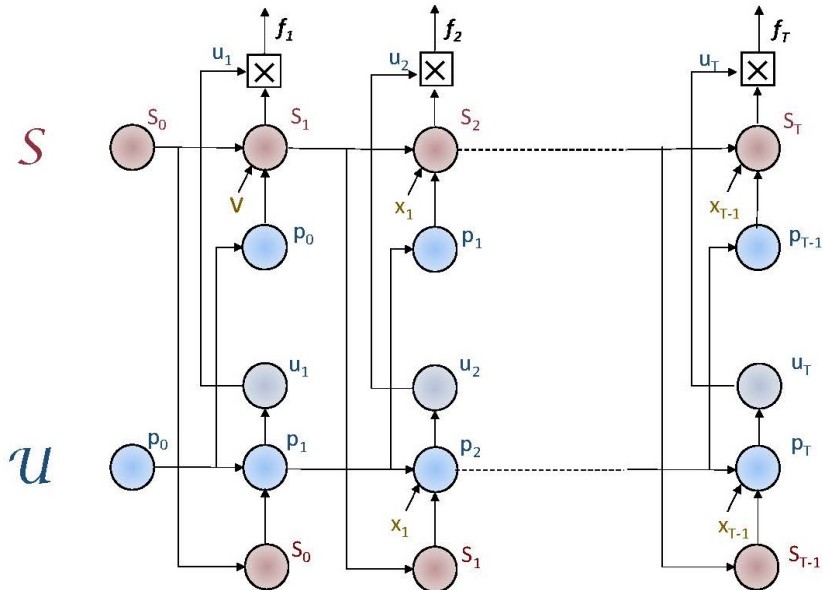

Figure 2: The sentence-encoding subnet $\mathcal{S}$ and the unbinding subnet $\mathcal{U}$ are inter-connected LSTMs; $\mathbf{v}$ encodes the visual input while the $\mathbf{x}_t$ encode the words of the output caption.

where $\mathbf{v} \in \mathbb{R}^{2048}$ is the vector of visual features extracted from the current image by ResNet (Gan et al. (2017)) and $\bar{\mathbf{v}}$ is the mean of all such vectors; $\mathbf{C}_s \in \mathbb{R}^{d \times d \times 2048}$. On the output side, $\mathbf{x}_t \in \mathbb{R}^V$ is a 1-hot vector with dimension equal to the size of the caption vocabulary, $V$, and $\mathbf{W}_e \in \mathbb{R}^{d \times V}$ is a word embedding matrix, the $i$-th column of which is the embedding vector of the $i$-th word in the vocabulary; it is obtained by the Stanford GLoVe algorithm with zero mean (Pennington et al. (2017)). $\mathbf{x}_0$ is initialized as the one-hot vector corresponding to a "start-of-sentence" symbol.

For $\mathcal{U}$ in Fig. 1, the state updating equations are:

$$\hat{\mathbf{f}}_{2,t} = \sigma_g(\hat{\mathbf{S}}_{t-1}\mathbf{w}_{2,f} - \mathbf{D}_{2,f}\mathbf{W}_e\mathbf{x}_{t-1} + \mathbf{U}_{2,f}\mathbf{p}_{t-1}) \tag{11}$$

$$\hat{\mathbf{i}}_{2,t} = \sigma_g(\hat{\mathbf{S}}_{t-1}\mathbf{w}_{2,i} - \mathbf{D}_{2,i}\mathbf{W}_e\mathbf{x}_{t-1} + \mathbf{U}_{2,i}\mathbf{p}_{t-1}) \tag{12}$$

$$\hat{\mathbf{o}}_{2,t} = \sigma_g(\hat{\mathbf{S}}_{t-1}\mathbf{w}_{2,o} - \mathbf{D}_{2,o}\mathbf{W}_e\mathbf{x}_{t-1} + \mathbf{U}_{2,o}\mathbf{p}_{t-1}) \tag{13}$$

$$\mathbf{g}_{2,t} = \sigma_h(\hat{\mathbf{S}}_{t-1}\mathbf{w}_{2,c} - \mathbf{D}_{2,c}\mathbf{W}_e\mathbf{x}_{t-1} + \mathbf{U}_{2,c}\mathbf{p}_{t-1}) \tag{14}$$

$$\mathbf{c}_{2,t} = \hat{\mathbf{f}}_{2,t} \odot \mathbf{c}_{2,t-1} + \hat{\mathbf{i}}_{2,t} \odot \mathbf{g}_{2,t} \tag{15}$$

$$\mathbf{p}_t = \hat{\mathbf{o}}_{2,t} \odot \sigma_h(\mathbf{c}_{2,t}) \tag{16}$$

where $\mathbf{w}_{2,f}, \mathbf{w}_{2,i}, \mathbf{w}_{2,o}, \mathbf{w}_{2,c} \in \mathbb{R}^d$, $\mathbf{D}_{2,f}, \mathbf{D}_{2,i}, \mathbf{D}_{2,o}, \mathbf{D}_{2,c} \in \mathbb{R}^{d \times d}$, and $\mathbf{U}_{2,f}, \mathbf{U}_{2,i}, \mathbf{U}_{2,o}, \mathbf{U}_{2,c} \in \mathbb{R}^{d \times d}$. The initial state $\mathbf{p}_0$ is the zero vector.

The dimensionality of the crucial vectors shown in Fig. 1, $\mathbf{u}_t$ and $\mathbf{f}_t$, is increased from $d \times 1$ to $d^2 \times 1$ as follows. A block-diagonal $d^2 \times d^2$ matrix $\mathbf{S}_t$ is created by placing $d$ copies of the $d \times d$ matrix $\hat{\mathbf{S}}_t$ as blocks along the principal diagonal. This matrix is the output of the sentence-encoding subnetwork $\mathcal{S}$. Now, following Eq. (3), the 'filler vector' $\mathbf{f}_t \in \mathbb{R}^{d^2}$ — 'unbound' from the sentence representation $\mathbf{S}_t$ with the 'unbinding vector' $\mathbf{u}_t$ — is obtained by Eq. (17).

$$\mathbf{f}_t = \mathbf{S}_t\mathbf{u}_t \tag{17}$$

Here $\mathbf{u}_t \in \mathbb{R}^{d^2}$, the output of the unbinding subnetwork $\mathcal{U}$, is computed as in Eq. (18), where $\mathbf{W}_u \in \mathbb{R}^{d^2 \times d}$ is $\mathcal{U}$'s output weight matrix.

$$\mathbf{u}_t = \sigma_h(\mathbf{W}_u\mathbf{p}_t) \tag{18}$$

Finally, the lexical subnetwork $\mathcal{L}$ produces a decoded word $\mathbf{x}_t \in \mathbb{R}^V$ by

$$\mathbf{x}_t = \sigma_s(\mathbf{W}_x\mathbf{f}_t) \tag{19}$$

where $\sigma_s(\cdot)$ is the softmax function and $\mathbf{W}_x \in \mathbb{R}^{V \times d^2}$ is the overall output weight matrix. Since $\mathbf{W}_x$ plays the role of a word de-embedding matrix, we can set

$$\mathbf{W}_x = (\mathbf{W}_e)^\top \tag{20}$$

where $\mathbf{W}_e$ is the word-embedding matrix. Since $\mathbf{W}_e$ is pre-defined, we directly set $\mathbf{W}_x$ by Eq. (20) without training $\mathcal{L}$ through Eq. (19). Note that $\mathcal{S}$ and $\mathcal{U}$ are learned jointly through the end-to-end training.

## 6 EXPERIMENTAL RESULTS

### 6.1 DATASET

To evaluate the performance of our proposed architecture, we use the COCO dataset (COCO (2017)). The COCO dataset contains 123,287 images, each of which is annotated with at least 5 captions. We use the same pre-defined splits as Karpathy & Fei-Fei (2015); Gan et al. (2017): 113,287 images for training, 5,000 images for validation, and 5,000 images for testing. We use the same vocabulary as that employed in Gan et al. (2017), which consists of 8,791 words.

### 6.2 EVALUATION OF IMAGE CAPTIONING SYSTEM

For the CNN of Fig. 1, we used ResNet-152 (He et al. (2016)), pretrained on the ImageNet dataset. The feature vector $\mathbf{v}$ has 2048 dimensions. Word embedding vectors in $\mathbf{W}_e$ are downloaded from the web (Pennington et al. (2017)). The model is implemented in TensorFlow (Abadi et al. (2015)) with the default settings for random initialization and optimization by backpropagation.

In our experiments, we choose $d = 25$ (where $d$ is the dimension of vector $\mathbf{p}_t$). The dimension of $\mathbf{S}_t$ is $625 \times 625$ (while $\hat{\mathbf{S}}_t$ is $25 \times 25$); the vocabulary size $V = 8,791$; the dimension of $\mathbf{u}_t$ and $\mathbf{f}_t$ is $d^2 = 625$.

Table 1: Performance of the proposed TPGN model on the COCO dataset.

| Methods | METEOR | BLEU-1 | BLEU-2 | BLEU-3 | BLEU-4 | CIDEr |
|---|---|---|---|---|---|---|
| NIC Vinyals et al. (2015) | – | 0.666 | 0.461 | 0.329 | 0.246 | – |
| CNN-LSTM | 0.238 | 0.698 | 0.525 | 0.390 | 0.292 | 0.889 |
| TPGN | **0.243** | **0.709** | **0.539** | **0.406** | **0.305** | **0.909** |

The main evaluation results on the MS COCO dataset are reported in Table 1. The widely-used BLEU (Papineni et al. (2002)), METEOR (Banerjee & Lavie (2005)), and CIDEr (Vedantam et al. (2015)) metrics are reported in our quantitative evaluation of the performance of the proposed schemes. In evaluation, our baseline is the widely used CNN-LSTM captioning method originally proposed in Vinyals et al. (2015). For comparison, we include results in that paper in the first line of Table 1. We also re-implemented the model using the latest ResNet feature and report the results in the second line of Table 1. Our re-implementation of the CNN-LSTM method matches the performance reported in Gan et al. (2017), showing that the baseline is a state-of-the-art implementation. As shown in Table 1, compared to the CNN-LSTM baseline, the proposed TPGN significantly outperforms the benchmark schemes in all metrics across the board. The improvement in BLEU-$n$ is greater for greater $n$; TPGN particularly improves generation of longer subsequences. The results clearly attest to the effectiveness of the TPGN architecture.

## 7 CONCLUSION

In this paper, we proposed a new Tensor Product Generation Network (TPGN) for natural language generation and related tasks. The model has a novel architecture based on a rationale derived from the use of Tensor Product Representations for encoding and processing symbolic structure through neural network computation. In evaluation, we tested the proposed model on captioning with the MS COCO dataset, a large-scale image captioning benchmark. Compared to widely adopted LSTM-based models, the proposed TPGN gives significant improvements on all major metrics including METEOR,

BLEU, and CIDEr. Moreover, we observe that the unbinding vectors contain important grammatical information. Our findings in this paper show great promise of TPRs. In the future, we will explore extending TPR to a variety of other NLP tasks.

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
