# OpenReview forum: "A Neural-Symbolic Approach to Natural Language Tasks"
_ICLR.cc/2018/Conference — Reject_

### Official Review · AnonReviewer1 · 2017-11-23
**Worthwhile goal of bringing together GSC and DL; really poor presentation.**

**Rating:** 4
**Confidence:** 5

**Review:**

I have a huge amount of sympathy for this work, and was really hoping to read a well-presented paper setting out how to cleanly integrate Smolensky's theory with deep learning, ideally (but not necessarily) with some decent empirical results. If that had been the case, I would certainly have been recommending acceptance, since ICLR would benefit from the alternative perspective that Smolensky's work provides, compared to the majority of work in Deep Learning. The empirical results are decent, but the presentation requires too much work to warrant acceptance at ICLR for this year. There are also some questionable decisions made regarding the NLP evaluations.

More detailed comments.

o Just call the pos tagging task "POS tagging". Talking about classification of the part of speech of *a* word makes it sound like you're tagging a single word in isolation.

o The statement that language structures can't be integrated with DL is a hopeless misrepresentation of current practice in NLP, and misses a large body of existing work. There's obviously Richard Socher's work on integrating DL and the output of eg the Stanford parser, but also a current raft of work on trying to induce tree structures automatically via a DL framework and task-based objective. Two examples, by no means exhaustive:

Learning to Compose Words into Sentences with Reinforcement Learning
Dani Yogatama, Phil Blunsom, Chris Dyer, Edward Grefenstette, Wang Ling.
ICLR 2017.

Jointly Learning Sentence Embeddings and Syntax with Unsupervised Tree-LSTMs
Jean Maillard, Stephen Clark, Dani Yogatama.

o It's not at all clear to me what the third task is - something like chunking. Identification of the phrase structure sounds like parsing, but you're not doing full parsing. This needs explaining fully. There are various standard NLP tasks related to identifying phrase structure - I would just do one of those, using one of the standard datasets, then there won't be any confusion.

o The description of tagging on p.2 mentions MEMMs too much - these were superseded by CRFs, which I think is what you mean to refer to.

o The reference to a maximum entropy language model is a little odd, since as far as I know these never became mainstream (assuming you mean Rosenfeld's whole sentence maxent language models).

o N is terrible name for a system!

o Not sure about the 5-role schema example on p.4, since presumably these would still be generated one word at a time? So in what sense is the model encoding a schema?

o Section 5 is the key section in the paper. Unfortunately I found it hard to follow. I guess the LSTM equations are needed for completeness, but what I really needed was a clear paragraph explaining how the LSTM is used to build the vectors and matrices used by the binding/unbinding network.

o There are various oddities in the POS tagging experiments. Why use the Stanford tagger? Are you using this to get the train/test data? Just use the Penn Treebank, or another standard pos tag dataset.

o Why precision and recall for tagging? Doesn't each word get assigned a single tag? In which case we just need accuracy.

o There are a number of minor comments I could have made re. the presentation, eg funny refs with both names (Chen and Laurence Zitnick).

---

> ### Author Response · Authors · 2018-01-05
> **Authors' reply to Reviewer 3**
>
>
> o Reviewer’s comment: Just call the pos tagging task "POS tagging". Talking about classification of the part of speech of *a* word makes it sound like you're tagging a single word in isolation.
>
>    Answer:  Thank you for the comment. This part has been removed.
>
> o Reviewer’s comment: The statement that language structures can't be integrated with DL is a hopeless misrepresentation of current practice in NLP, and misses a large body of existing work. There's obviously Richard Socher's work on integrating DL and the output of eg the Stanford parser, but also a current raft of work on trying to induce tree structures automatically via a DL framework and task-based objective. Two examples, by no means exhaustive: Learning to Compose Words into Sentences with Reinforcement Learning Dani Yogatama, Phil Blunsom, Chris Dyer, Edward Grefenstette, Wang Ling. ICLR 2017. Jointly Learning Sentence Embeddings and Syntax with Unsupervised Tree-LSTMs Jean Maillard, Stephen Clark, Dani Yogatama.
>
>     Answer: Thank you for your comments.  Our grammatical interpretation of the structural roles of words in sentences makes contact with other work that incorporates deep learning into grammatically structured networks as mentioned by the reviewer. In our paper, the network
> is not itself structured to match the grammatical structure of sentences being processed; the structure is fixed, but is designed to support the learning of distributed representations that incorporate structure internal to the representations themselves — filler/role structure.
>
> o Reviewer’s comment: It's not at all clear to me what the third task is - something like chunking. Identification of the phrase structure sounds like parsing, but you're not doing full parsing. This needs explaining fully. There are various standard NLP tasks related to identifying phrase structure - I would just do one of those, using one of the standard datasets, then there won't be any confusion.
>
>    Answer: Thank you for the comment.  We will conduct more comprehensive study and present the findings in future work.
>
> o Reviewer’s comment: The description of tagging on p.2 mentions MEMMs too much - these were superseded by CRFs, which I think is what you mean to refer to.
>
>    Answer: Yes, you are right.
>
> o Reviewer’s comment: The reference to a maximum entropy language model is a little odd, since as far as I know these never became mainstream (assuming you mean Rosenfeld's whole sentence maxent language models).
>
>    Answer: In the literature, the maximum entropy approach is one type of language model as mentioned in the following paper:
> Devlin, Jacob, Hao Cheng, Hao Fang, Saurabh Gupta, Li Deng, Xiaodong He, Geoffrey Zweig, and Margaret Mitchell. "Language models for image captioning: The quirks and what works." arXiv preprint arXiv:1505.01809 (2015).
>
> o Reviewer’s comment: N is terrible name for a system!
>
>     Answer: It is a notation for the TPGN system.  This notation is from "N" in "TPGN".
>
> o  Reviewer’s comment: Not sure about the 5-role schema example on p.4, since presumably these would still be generated one word at a time? So in what sense is the model encoding a schema?
>
>     Answer: 5-role schema corresponds to five role vectors r_1, r_2, r_3, r_4, r_5.  The model encodes 5-role schema with fillers by Eq. (1).
>
> o Reviewer’s comment: Section 5 is the key section in the paper. Unfortunately I found it hard to follow. I guess the LSTM equations are needed for completeness, but what I really needed was a clear paragraph explaining how the LSTM is used to build the vectors and matrices used by the binding/unbinding network.
>
>     Answer: Thank you for your comment. We re-organized the paper to improve its clarity.
>
> o Reviewer’s comment: There are various oddities in the POS tagging experiments. Why use the Stanford tagger? Are you using this to get the train/test data? Just use the Penn Treebank, or another standard pos tag dataset.
>
>     Answer: Thank you for good suggestion. we will perform more comprehensive study and present the results in future work.
>
> o Reviewer’s comment: Why precision and recall for tagging? Doesn't each word get assigned a single tag? In which case we just need accuracy.
>
>    Answer: Thanks. we will perform more comprehensive study and present the results in future work.
>
> o Reviewer’s comment:  There are a number of minor comments I could have made re. the presentation, eg funny refs with both names (Chen and Laurence Zitnick).
>
>     Answer: We fixed it in the revised paper.

---

### Official Review · AnonReviewer3 · 2017-11-27
**The paper adapts a classical idea of Tensor Product Representation (TPN) which basically talks about how connectionist models can capture syntax explicitly and adapts the TPN as an inductive bias in image caption generation, and shows the generality of the learnt representations to tasks such as POS tagging and phrase classification.**

**Rating:** 5
**Confidence:** 4

**Review:**

**Strengths**
The approach to sentence generation makes a lot of sense -- and provides a potentially elegant manner to incorporate or provide the model with the inductive bias that language has syntax and semantics, by leveraging a classical idea called Tensor Product Representation and showing how to adapt it to modern deep learning architectures and “learn” syntax and semantics end to end. Results on image captioning models indicate that the proposed approach might be promising. As a by-product, the paper also evaluates the model on POS tagging and shows that one can do fairly well using the representations learned in the TPGN.

**Weakness**
My main concerns are the lack of appropriate baselines to establish concretely the contribution of the TPGN. It would be good to address issues under “Baselines” below.

Approach:
1. It would be nice to explain clearly why the pretraining of the TPGN with the LSTM input is needed. Is the idea that one would want to feed the representation of the entire representation as input in order to infer what “S_n” should be? Why is it then justified to feed in the image input instead? Also, in the second stage, the image features need not correspond to the LSTM feature dimensions, which means that the pretraining seems unprincipled. A better solution would have been to learn a joint embedding of image captions and labels (say via. ranking), and then use the embedding for the caption as input to the TPGN. This would ensure that when we use images, the model sees input that is appropriately “aligned”. A discussion why this is not needed or implementing this seems important.

Minor Points:
1.  “There are mainly two approaches to natural language generation in image captioning. The first approach takes the words detected by a CNN as input, and uses a probabilistic model, such as a maximum entropy (ME) language model, to arrange the detected words into a sentence.” -- can cite Fang. et.al [A]

Baselines:
1. The arXiv version and the PAMI version of the Neural Image Captioning paper (Vinyals, 2015) does report numbers on METEOR and CIDEr metrics, so they should be used to populate Table. 1 for completeness. Also, it would be good to clarify which split of MSCOCO Table. 1 reports results on -- is it the 40K large validation split or the 5K validation/test split released by (Karpathy, 2015)? Clarifying this would be nice since the numbers seem a bit on the lower side.

2. What are the relative number of parameters in the Vinyals et.al. baseline and the proposed TPGN model? Would having an LSTM with twice the number of layers (by stacking them) or twice the size of the hidden state do better?

3. What if we used the hidden state of a regular LSTM decoder to do POS tagging? How well would that do? Does the TPN capture any more syntactic structure than an LSTM decoder (Table. 2). This seems to be an important result to report.

Clarity:
1. Page 7.: “We also implemented the latest ResNet feature” -- would be good be explicit which resnet model is used.

References:
[A]: Fang, Hao, Saurabh Gupta, Forrest Iandola, Rupesh Srivastava, Li Deng, Piotr Dollár, Jianfeng Gao, et al. 2014. “From Captions to Visual Concepts and Back.” arXiv [cs.CV]. arXiv. http://arxiv.org/abs/1411.4952.

---

> ### Author Response · Authors · 2018-01-05
> **Authors' reply to Reviewer 2**
>
>
> I) Approach:
> 1) Reviewer’s comment: It would be nice to explain clearly why the pretraining of the TPGN with the LSTM input is needed. Is the idea that one would want to feed the representation of the entire representation as input in order to infer what “S_n” should be? Why is it then justified to feed in the image input instead? Also, in the second stage, the image features need not correspond to the LSTM feature dimensions, which means that the pretraining seems unprincipled. A better solution would have been to learn a joint embedding of image captions and labels (say via. ranking), and then use the embedding for the caption as input to the TPGN. This would ensure that when we use images, the model sees input that is appropriately “aligned”. A discussion why this is not needed or implementing this seems important.
>
>     Answer:  Thanks for the suggestion.  We removed it from the paper.
>
> II) Minor Points:
> 1) Reviewer’s comment: “There are mainly two approaches to natural language generation in image captioning. The first approach takes the words detected by a CNN as input, and uses a probabilistic model, such as a maximum entropy (ME) language model, to arrange the detected words into a sentence.” -- can cite Fang. et.al [A]
>
>     Answer: Thank you for the suggestion.  We cited it in the revised paper.
>
> III) Baselines:
> 1) Reviewer’s comment: 1. The arXiv version and the PAMI version of the Neural Image Captioning paper (Vinyals, 2015) does report numbers on METEOR and CIDEr metrics, so they should be used to populate Table. 1 for completeness. Also, it would be good to clarify which split of MSCOCO Table. 1 reports results on -- is it the 40K large validation split or the 5K validation/test split released by (Karpathy, 2015)? Clarifying this would be nice since the numbers seem a bit on the lower side.
>
>     Answer: Thank you for pointing it out.  We corrected the numbers in the revised paper.
>
> 2) Reviewer’s comment: What are the relative number of parameters in the Vinyals et.al. baseline and the proposed TPGN model? Would having an LSTM with twice the number of layers (by stacking them) or twice the size of the hidden state do better?
>
>     Answer: The complexity of the TPGN is comparable to the NIC. The results from NIC and our re-implementation in the paper are optimized w.r.t. the model architecture on hold-out data, and we observed that the performance start to saturate when adding more layers/nodes in the CNN-LSTM re-implementation.
>
> 3) Reviewer’s comment: What if we used the hidden state of a regular LSTM decoder to do POS tagging? How well would that do? Does the TPN capture any more syntactic structure than an LSTM decoder (Table. 2). This seems to be an important result to report.
>
>     Answer: The accuracy of using the hidden state of a regular LSTM decoder to do POS tagging is less than 70%, which is much poorer than the Stanford POS tagger and our TPR based POS tagger.   Thanks for the suggestion. We will perform this experiments in future work.
>
> IV) Clarity:
> I) Reviewer’s comment:  Page 7.: “We also implemented the latest ResNet feature” -- would be good be explicit which resnet model is used.
>
>    Answer: We take the output of the 2048-way pool5 layer from ResNet-152, pretrained on the
> ImageNet dataset. This feature is used in the CNN-LSTM reimplementation and our TPGN.

---

### Official Review · AnonReviewer2 · 2017-11-30
**Confusing paper, poor experiments and missing a lot of references.**

**Rating:** 4
**Confidence:** 4

**Review:**

The paper claims that "Deep Learning (DL) has not been able to explicitly represent and enforce grammatical structures", which is false, see "Improved Semantic Representations From Tree-Structured Long Short-Term Memory Networks", "Ask Me Anything: Dynamic Memory Networks for Natural Language Processing", "DRAGNN: A Transition-based Framework for Dynamically Connected Neural Networks" or "Deep Compositional Question Answering with Neural Module Networks".

The Introduction triple challenge is confusing, not clear what are the challenges this paper tries to address.

"The representation learned in a crucial layer of the TPGN can be interpreted as encoding grammatical roles" Doesn't refer to any specific kind of layer, or what it make it special.

The idea of using outer product as a layer has already been explored in "Multimodal compact bilinear pooling for visual question answering and visual grounding"

The following paragraph in page 2 is not clear, very confusing:
The work reported here .... their categories


In page 3 authors claim that the "vectors are linearly independent" but didn't specify how they enforce that.

Figure 3 contradicts Figure 1, not clear what are the inputs for module S.

The experiments reported in Table1 are useless, there a tons of previous work with much better results, see
https://competitions.codalab.org/competitions/3221#results

Even the numbers reported for Vinyals et al. (2015) are much higher in the leaderboard.

There is no comparison with other models that use attention or analysis of the impact of the increased number of parameters of the method proposed.

The experiments about POS tagger and Phrase Classifier are reported on 5000 from the COCO test set, which is useful for comparisons. Should report numbers on PennTreeBank or other common POS dataset.

The text is missing a lot of references, for example:
 - page 2 GSC
 - page 2 The first approach takes the detected by a CNN ....
 - page 3 previous work where TPRs are hand-crafted

---

### Decision · Program_Chairs · 2018-01-29
**ICLR 2018 Conference Acceptance Decision**

**Decision:**

Reject

**Comment:**

This work presents a neuro-symbolic model that hopes to encode structural aspects of language into hidden space representations. The reviewers were very sympathetic to the motivation of the work, but felt that it was lacking in two significant areas:

- Baselines and empirical comparisons on the tasks presented. In particular the use of state-of-the-art systems and "previous work with much better results"
- Related work on the problem of incorporating structural aspects of language into neural systems, in particular work that uses tree structured neural systems.